# Remote transport of high-dimensional orbital angular momentum states and ghost images via spatial-mode-engineered frequency conversion

Xiaodong Qiu[1], Haoxu Guo[1] & Lixiang Chen [1] ✉

The efficient transport and engineering of photonic orbital angular momentum (OAM) lie at the heart of various related classical and quantum applications. Here, by leveraging the spatial-mode-engineered frequency conversion, we realize the remote transport of high-dimensional orbital angular momentum (OAM) states between two distant parties without direct transmission of information carriers. We exploit perfect vortices for preparing high-dimensional yet maximal O AM entanglement. Based on nonlinear sum-frequency generation working with a strong coherent wave packet and a single photon, we conduct the Bell-like state measurements for high-dimensional perfect vortices. We experimentally achieve an average transport fidelity $0.879 \pm 0.048$ and $0.796 \pm 0.066$ for a complete set of 3-dimensional and 5-dimensional OAM mutually unbiased bases, respectively. Furthermore, by exploring the full transverse entanglement, we construct another strategy of quantum imaging with interaction-free light. It is expected that, with the future advances in nonlinear frequency conversion, our scheme will pave the way for realizing truly secure high-dimensional quantum teleportation in the upcoming quantum network.

Since the seminal discovery by Allen et al.[1], the high-dimensional nature of light's orbital angular momentum (OAM) has attracted significant attention due to its potential in some classical and quantum information protocols[2–5], e.g., realizing large-alphabet optical communication[6,7] and constructing the high-dimensional photonic quantum entanglement[8], thus, providing the toolkit for the investigation of quantum fundamental questions[9]. However, these benefits are only conceivable when we are able to efficiently generate, manipulate, transport, and detect the OAM state. Hitherto, extensive investigations of OAM transport[6,7,10–17] have been reported, such as OAM transport in free-space, specialized optical fiber, and even the distribution of OAM entanglement. Whereas, we note that most of these previous schemes focused only on the direct information transport.

Recent years have also witnessed a growing interest in the nonlinear frequency conversion of structured light[5,18]. In spatial domain, it was demonstrated that using OAM light as fundamental frequency light source, one can realize the image processing and mode detection during the process of upconversion imaging[19,20], and by shaping the spatial shape of the pump source, one can enhance the performance of high-dimensional OAM frequency interface[21–24]. In time-frequency domain, by using the dispersion-engineered sum-frequency generation, the quantum pulse gate for harnessing and selecting the time-frequency mode has been realized[25,26]. Also, sum-frequency generation in conjunction with pulse-shaping techniques provided a powerful tool for time-bin entanglement measurement[27]. Particularly, we note that there have been several attempts, by means of sum-frequency generation, to perform the coherent transfer of the

[1]Department of Physics, Xiamen University, Xiamen 361005, China. ✉e-mail: chenlx@xmu.edu.cn

polarization state between two distant parties without direct transmission of the information[28], and to realize the entanglement swapping of time-bin entanglement[29,30], however, the scenario of high-dimensional OAM state has not yet been fully explored. Some interesting questions arise naturally: whether the nonlinear frequency conversion can be used to transport the high-dimensional OAM state in an unprecedented way, and what features this might lead to. These also form the major incentive of our present work.

Here, by leveraging the spatial-mode-engineered frequency conversion, we realize the remote transport of high-dimensional orbital angular momentum (OAM) states at a distance without direct transmission of information carriers. Specifically, by exploiting perfect vortices, we prepare high-dimensional yet maximally entangled orbital angular momentum (OAM) states as communication channel. Then, we employ sum-frequency generation (SFG) with a strong coherent state, which is used to bear the high-dimensional OAM state, to enhance the frequency conversion, and thus performing the perfect-vortex-based high-dimensional Bell-like state measurement reliably. Furthermore, we report the quantum imaging with truly interaction-free light.

## Results

### Remote transport of high-dimensional OAM states: model and implementation

Twisted photons carrying OAM possesses an inherent capacity for constructing a high-dimensional Hilbert space, thus making it a promising candidate for high-dimensional quantum information processing. It was Allen and coworkers[1] that recognized that the Laguerre-Gaussian (LG) modes of helical phase $\exp(i\ell\phi)$ carries a well-defined OAM of $\ell\hbar$. Generally, two-photon OAM-entangled state generated via spontaneous parametric down-conversion (SPDC) with a fundamental Gaussian pump $LG_0^0(r,\phi)$ can be written as, $|\Psi\rangle = \sum_\ell C_{\ell,-\ell}|\ell\rangle_s|-\ell\rangle_i$, where $C_{\ell,-\ell}$ denotes the probability amplitude of finding the signal photon in the standard LG mode $LG_{p=0}^\ell$ (often abbreviated to $|\ell\rangle_s$) and the idler one in $LG_{p=0}^{-\ell}$ ($|\ell\rangle_i$). Specifically, we have $C_{\ell,-\ell} = \int LG_0^0(r,\phi)[LG_0^\ell(r,\phi)]^*[LG_0^{-\ell}(r,\phi)]^*$. As for standard LG modes, the radius of the doughnut-like patterns scale with $\sqrt{\ell}$, a calculation finds that $C_{\ell,-\ell}$ will decrease as $|\ell|$ increases, resulting in a non-maximally entangled OAM state[31], $|\Psi\rangle$. However, the maximal entanglement is prerequisite for our remote transport of high-dimensional OAM states. The Procrustean method of entanglement concertation was proposed to enhance the entanglement. Here, we employ the perfect vortices[32] for the same purpose. We modify the LG modes as, $MLG_0^\ell(r,\phi) = |LG_0^l(r,\phi)|\exp(i\ell\phi)$ (with $l$ being a constant, e.g., $l=5$), to make different OAM modes yet bear the same radial intensity distribution, i.e., perfect vortices (Methods). Then, according to $C_{\ell,-\ell} = \int LG_0^0(r,\phi)[MLG_0^\ell(r,\phi)]^*[MLG_0^{-\ell}(r,\phi)]^*$, we know that $C_{\ell,-\ell}$ can naturally be equalized because of the identical overlap probability regardless of different $\ell$. Besides, it is noted that these modified modes still form an orthogonal and complete basis. In other words, our adoption of perfect vortices enables the additional exploration of radial correlation and ensures the identical two-photon mode amplitudes, thus maximizing the high-dimensional OAM entanglement. Without loss of generality, we first consider the transport of 3-dimensional OAM states. Assume Alice wishes to transport to Bob the OAM state, $|\varphi\rangle_a = \alpha_1|-1\rangle_a + \alpha_2|0\rangle_a + \alpha_3|1\rangle_a$, which is carried by a strong coherent light $a$, and the complex coefficients $\alpha_1$, $\alpha_2$, and $\alpha_3$ fulfill $|\alpha_1|^2 + |\alpha_2|^2 + |\alpha_3|^2 = 1$. For transporting such arbitrary 3-dimensional states, Alice and Bob need to share the aforementioned high-dimensional and maximally entangled state (photons $b$ and $c$),

$$|\psi\rangle_{bc} = \frac{1}{\sqrt{3}}(|-1,1\rangle_{bc} + |0,0\rangle_{bc} + |1,-1\rangle_{bc}) \quad (1)$$

Then, Alice mixes the coherent light $a$ and photon $b$ in the nonlinear crystal to conduct the sum-frequency generation (SFG). Owing to the OAM conservation in SFG[33], if we detect the SFG photon with a single-mode fiber (SMF), which can only support fundamental mode, and thus, the topological charges of coherent light $a$ and photon $b$ satisfy the relation, $\ell_a + \ell_b = 0$. Also, owing to the OAM conservation in SPDC, seen in Eq. (1), the coincidence between SFG photon and photon $c$ can only be detected when $\ell_a = -\ell_b = \ell_c$. Accordingly, the OAM state of the coherent light $a$ could be transported to photon $c$ without direct transmission of information carriers. However, due to the non-uniform frequency conversion efficiency for different input OAM eigenstates, e.g., the LG modes, the transport performance will get worse with the increase of OAM value. Here, we adopt the perfect vortices to overcome this obstacle, as they can guarantee the identical conversion efficiency for different pairwise OAM modes $|\ell\rangle_a|-\ell\rangle_b$. Thus, the high-dimensional OAM states can be transported reliably (Methods).

In other words, after nonlinear projection via SMF, the OAM states of coherent light $a$ and converted photon $b$ satisfy $|\psi\rangle_{ab} = \frac{1}{\sqrt{3}}(|-1,1\rangle_{ab} + |0,0\rangle_{ab} + |1,-1\rangle_{ab})$. It's noted that, due to the coherent light $a$ adopted here, $|\psi\rangle_{ab}$ is just used to characterize the dependence relationship between $a$ and $b$, and does not represent a regular entanglement state. In other words, it can be seen as a hybrid classical and quantum system. Interestingly, the relationship $|\psi\rangle_{ab}$ is formally similar with one of the 3-dimensional Bell states, $|\psi_{mn}\rangle = \frac{1}{\sqrt{3}}\sum_{\ell=-1}^1 \exp(i2n\ell\pi/3)|\ell\rangle|-\ell\oplus m\rangle$ with $m,n \in \{0,1,2\}$. Thus here, we termed it as the high-dimensional Bell-like measurement. It is noted that the nonlinear SFG process has been employed to measure four complete Bell states for teleportation of a polarization state[28] and for time-energy entanglement swapping[29,30]. Inspiringly, if we perform full nine 3-dimensional Bell-like measurements by selecting the well-designed OAM modes of SFG photons, we can transport high-dimensional OAM state of coherent light $a$ to Bob and implement a corresponding unitary transform simultaneously, that is,

$$|\psi_{00}\rangle_{ab} \rightarrow |\varphi\rangle_c = \alpha_1|-1\rangle_c + \alpha_2|0\rangle_c + \alpha_3|1\rangle_c, \quad (2)$$

$$|\psi_{01}\rangle_{ab} \rightarrow |\varphi\rangle_c = \alpha_1\omega|-1\rangle_c + \alpha_2|0\rangle_c + \alpha_3\omega^2|1\rangle_c, \quad (3)$$

$$|\psi_{02}\rangle_{ab} \rightarrow |\varphi\rangle_c = \alpha_1\omega^2|-1\rangle_c + \alpha_2|0\rangle_c + \alpha_3\omega|1\rangle_c, \quad (4)$$

$$|\psi_{10}\rangle_{ab} \rightarrow |\varphi\rangle_c = \alpha_1|1\rangle_c + \alpha_2|-1\rangle_c + \alpha_3|0\rangle_c, \quad (5)$$

$$|\psi_{11}\rangle_{ab} \rightarrow |\varphi\rangle_c = \alpha_1\omega|1\rangle_c + \alpha_2|-1\rangle_c + \alpha_3\omega^2|0\rangle_c, \quad (6)$$

$$|\psi_{12}\rangle_{ab} \rightarrow |\varphi\rangle_c = \alpha_1\omega^2|1\rangle_c + \alpha_2|-1\rangle_c + \alpha_3\omega|0\rangle_c, \quad (7)$$

$$|\psi_{20}\rangle_{ab} \rightarrow |\varphi\rangle_c = \alpha_1|0\rangle_c + \alpha_2|1\rangle_c + \alpha_3|-1\rangle_c, \quad (8)$$

$$|\psi_{21}\rangle_{ab} \rightarrow |\varphi\rangle_c = \alpha_1\omega|0\rangle_c + \alpha_2|1\rangle_c + \alpha_3\omega^2|-1\rangle_c, \quad (9)$$

$$|\psi_{22}\rangle_{ab} \rightarrow |\varphi\rangle_c = \alpha_1\omega^2|0\rangle_c + \alpha_2|1\rangle_c + \alpha_3\omega|-1\rangle_c, \quad (10)$$

with $\omega = \exp(i2\pi/3)$.

By a careful analysis on the OAM conversion in SFG process, we find that the desired 9 Bell-like states can be equivalently identified. First, Alice can mix coherent light $a$ and photon $b$ in the nonlinear crystal. Owing to OAM conservation in SFG[33], we can map the following 7 OAM Bell-like states onto 7 single-photon OAM superposition states, respectively, that is,

$$|\psi_{00}\rangle_{ab} \rightarrow |0\rangle, \tag{11}$$

$$|\psi_{10}\rangle_{ab} \rightarrow |-2\rangle + 2|1\rangle, \tag{12}$$

$$|\psi_{11}\rangle_{ab} \rightarrow \omega^2|-2\rangle + (\omega+1)|1\rangle, \tag{13}$$

$$|\psi_{12}\rangle_{ab} \rightarrow \omega|-2\rangle + (\omega^2+1)|1\rangle, \tag{14}$$

$$|\psi_{20}\rangle_{ab} \rightarrow |2\rangle + 2|-1\rangle, \tag{15}$$

$$|\psi_{21}\rangle_{ab} \rightarrow (\omega^2+1)|-1\rangle + \omega|2\rangle, \tag{16}$$

$$|\psi_{22}\rangle_{ab} \rightarrow (\omega+1)|-1\rangle + \omega^2|2\rangle. \tag{17}$$

Of interest is that, for $|\psi_{00}\rangle_{ab}$, the resultant SFG OAM state is orthogonal to the rest of Bell-like states. While for $|\psi_{01}\rangle_{ab}$ and $|\psi_{02}\rangle_{ab}$, the SFG for coherent light $a$ and photon $b$ is forbidden, as a result of the destructive interference. Still, we can identify $|\psi_{01}\rangle_{ab}$ and $|\psi_{02}\rangle_{ab}$ by performing a prior OAM-dependent phase shifts on coherent light $a$. This can be done simply by inserting into coherent light $a$'s path one Dove prism (DP)[34], whose orientation is adjusted at an angle of $-\pi/3$ and $\pi/3$, respectively, to convert $|\psi_{01}\rangle_{ab}$ and $|\psi_{02}\rangle_{ab}$ to $|\psi_{00}\rangle_{ab}$. By projecting the SFG photons onto $|-2\rangle + |1\rangle$ and $|2\rangle + |-1\rangle$, we can also identify $|\psi_{10}\rangle_{ab}$ and $|\psi_{20}\rangle_{ab}$, respectively. Similarly, by performing a

prior OAM-dependent phase shifts on coherent light $a$, we can also convert $|\psi_{11}\rangle_{ab}$ and $|\psi_{12}\rangle_{ab}$ to $|\psi_{10}\rangle_{ab}$, $|\psi_{21}\rangle_{ab}$ and $|\psi_{22}\rangle_{ab}$ to $|\psi_{20}\rangle_{ab}$, respectively. Thus all 9 Bell-like states can be identified in theory. In our proof-of-principle experimental demonstration, without loss of generality, we have identified three of all the high-dimensional OAM Bell-like states, respectively. In other words, the transport of high-dimensional OAM states could be realized by the perfect-vortex-encoded and SFG-based high-dimensional Bell-like state measurement (HDBLSM) in the OAM subspaces. Whereas, limited by the low efficiency of SFG at the single-photon level[28–30], if we use the single photon to replace the coherent light $a$ as input, we need to wait a much longer time for information transport. It is expected that, with the future advances in nonlinear frequency conversion, our scheme could be used to perform the truly high-dimensional Bell state measurement, and thus paves the way for realizing the secure high-dimensional quantum teleportation[35–39].

Figure 1 illustrate our experimental setup for remote transport of high-dimensional OAM states and the quantum imaging with interaction-free light. The 710 nm laser beam is directed to the Lithium triborate crystal (LBO), via second-harmonic generation, to generate the ultraviolet pulses centered at 355 nm. The residual 710 nm light is guided onto a spatial light modulator (SLM1), that displays suitable holograms, for preparing coherent beam $a$ in the desired high-dimensional OAM superpositions to be transported. While the generated 355 nm ultraviolet pulses are directed to pump the β-barium borate crystal (BBO1) to create the non-degenerate OAM-entangled photon pairs via SPDC, i.e., photon $b$ and $c$, centered at 780 nm and 650 nm, respectively. Alice possesses coherent beam $a$ and photon $b$ while Bob holds photon $c$. In our experiment without any prior concentration, we have achieved the preparation of 3-dimensional and 5-dimensional entangled states with a high fidelity of 96.7% and 89.5%, respectively (see Supplementary Note 1 for details). Then, Alice performs the HDBLSM on her coherent beam $a$ and photon $b$, by sending them together to BBO2 to do SFG. The generated 372 nm SFG photon

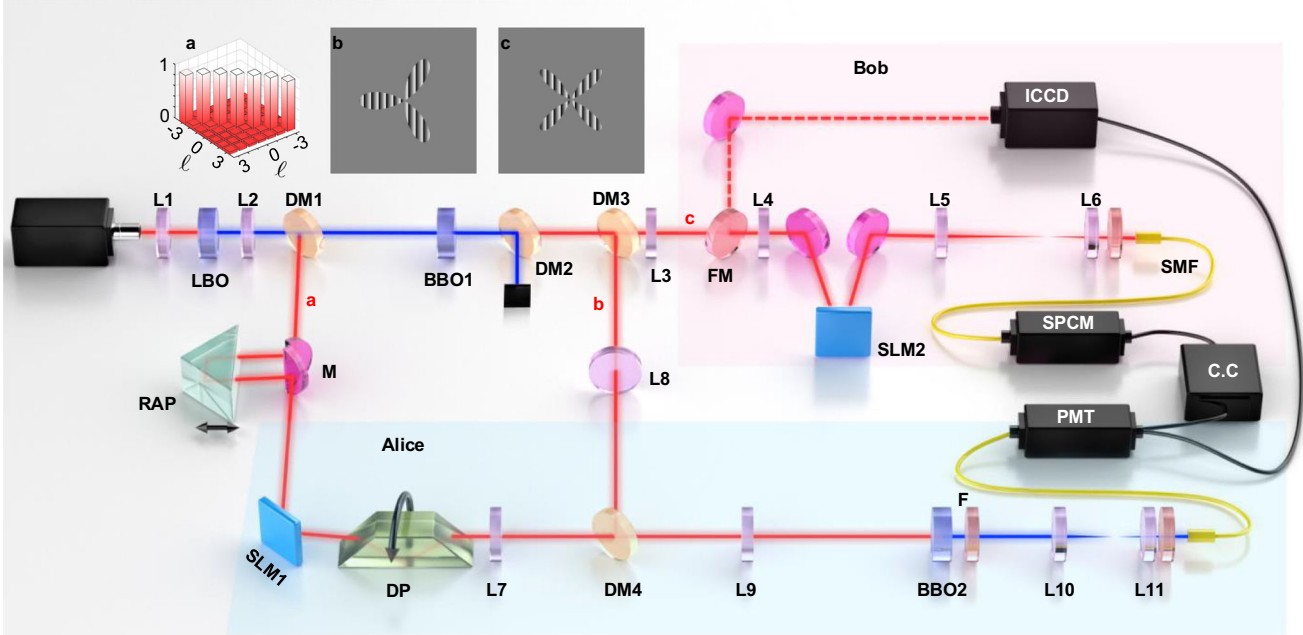

**Fig. 1 | Experimental setup for remote transport of high-dimensional OAM states and the quantum imaging with interaction-free light.** Inset **a** shows the measured two-photon 7-dimensional OAM spectrum, **b** and **c** illustrate the holographic gratings used for encoding the three-leaf and four-leaf Clover images. By switching the flipper mirror (FM), full tomography for the transported OAM superposition states and the capture of the ghost images can be conducted by using the coincidence circuit (C.C) and the ICCD camera, respectively. See the Methods for details. LBO Lithium triborate crystal, BBO β-barium borate crystal, DM dichroic mirror, SM single-mode fiber, SLM spatial light modulator, SPCM single photon counting module, PMT photon multiplier tube, RAP right-angle prism, DP Dove prism, M mirror, L lens, F filter.

is then coupled to a single-mode fiber (SMF) and detected by a photon multiplier tube (PMT). Thus the single-photon event from PMT at Alice's side indicates the successful HDBLSM and heralds a transferred photon $c$ at Bob's side.

### Experimental observations for 3- and 5-dimensional OAM states transport

In the first set of experiment, we prepare 12 OAM states to be transported at Alice's side, $|\varphi_1\rangle_a = |-1\rangle_a$, $|\varphi_2\rangle_a = |0\rangle_a$, $|\varphi_3\rangle_a = |1\rangle_a$, $|\varphi_4\rangle_a = (|-1\rangle_a + |0\rangle_a + |1\rangle_a)/\sqrt{3}$, $|\varphi_5\rangle_a = (|-1\rangle_a + \omega|0\rangle_a + \omega^2|1\rangle_a)/\sqrt{3}$, $|\varphi_6\rangle_a = (|-1\rangle_a + \omega^2|0\rangle_a + \omega|1\rangle_a)/\sqrt{3}$, $|\varphi_7\rangle_a = (\omega|-1\rangle_a + |0\rangle_a + |1\rangle_a)/\sqrt{3}$, $|\varphi_8\rangle_a = (|-1\rangle_a + \omega|0\rangle_a + |1\rangle_a)/\sqrt{3}$, $|\varphi_9\rangle_a = (|-1\rangle_a + |0\rangle_a + \omega|1\rangle_a)/\sqrt{3}$, $|\varphi_{10}\rangle_a = (\omega^2|-1\rangle_a + |0\rangle_a + |1\rangle_a)/\sqrt{3}$, $|\varphi_{11}\rangle_a = (|-1\rangle_a + \omega^2|0\rangle_a + |1\rangle_a)/\sqrt{3}$, and $|\varphi_{12}\rangle_a = (|-1\rangle_a + |0\rangle_a + \omega^2|1\rangle_a)/\sqrt{3}$. They constitute the full four sets of 3-dimensional mutually unbiased bases (MUB), which are

commonly used to testify the universality of transport for all possible superposition states[40]. The verification of transport results is based on the two-fold coincidence detections between the SFG photon (detected by PMT) and photon $c$ (detected by SPCM). After transporting, we employ the generalized Gell-Mann matrix basis[41] to reconstruct the density matrices of photon $c$ at Bob's side. Then the performance can be characterized by calculating the transport fidelity, $F = \text{Tr}(\rho_E \rho_T) + \sqrt{1 - \text{Tr}(\rho_E^2)}\sqrt{1 - \text{Tr}(\rho_T^2)}$, where $\rho_E$ and $\rho_T$ represent the normalized density matrices of the transported states and the original ones, respectively. We present in Fig. 2 the transport results for a qutrit OAM state, $|\varphi_4\rangle_a = (|-1\rangle_a + |0\rangle_a + |1\rangle_a)/\sqrt{3}$, without loss of generality, conditional on three different specific HDBLSM. Specifically, Fig. 2a, b, c show the resultant density matrices conditional on HDBLSM results of $|\psi_{00}\rangle_{ab}$, $|\psi_{01}\rangle_{ab}$, and $|\psi_{02}\rangle_{ab}$, respectively, in each of which the upper rows are the original real and imaginary parts while the bottom are the

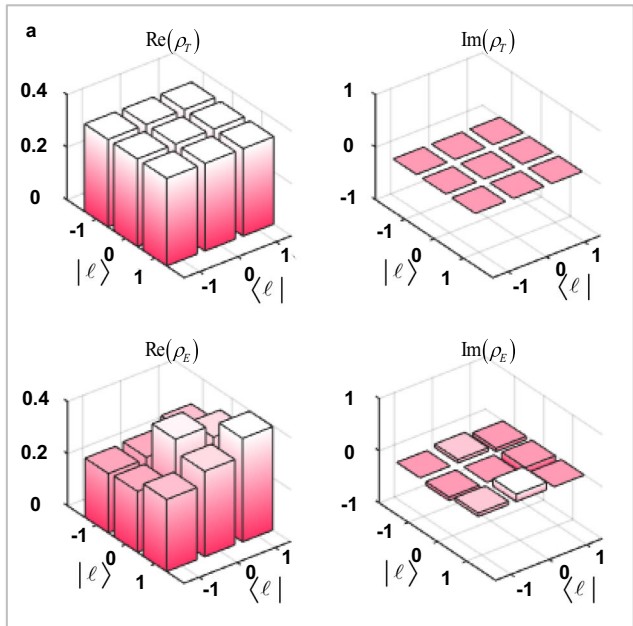

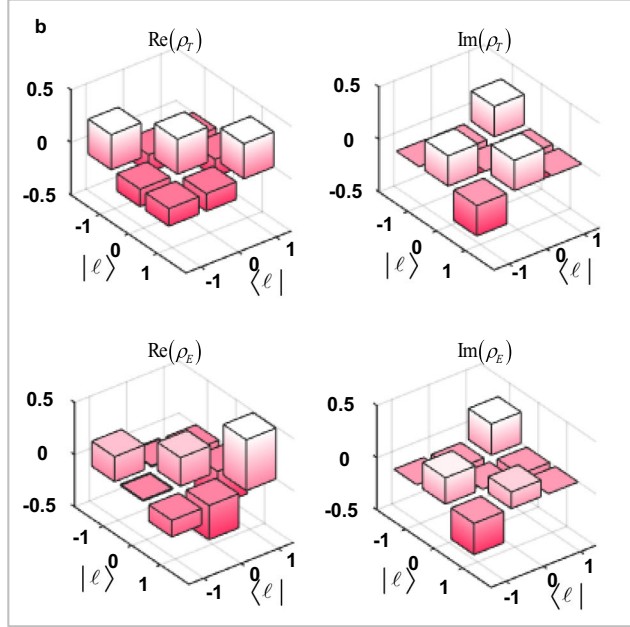

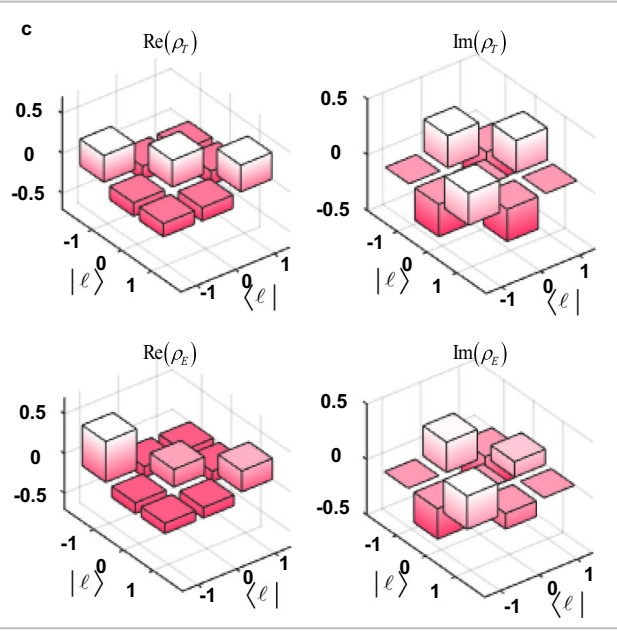

**Fig. 2 | Density matrices for transport of 3-dimensional OAM state, $|\varphi_4\rangle_a = (|-1\rangle_a + |0\rangle_a + |1\rangle_a)/\sqrt{3}$, conditional on three specific HDBLSMs. a** $|\psi_{00}\rangle_{ab}$ **b** $|\psi_{01}\rangle_{ab}$ and **c** $|\psi_{02}\rangle_{ab}$. In each case, the upper rows are the original real and imaginary parts while the bottom are the experimental results.

experimentally reconstructed ones. The fairly good agreement between them can be seen clearly. For quantitative analysis, we also show in Fig. 3 the transport fidelities for all 12 MUB states conditional on these three HDBLSM. The average transport fidelity is $0.879 \pm 0.048$.

Secondly, we conduct the transport of 5-dimensional OAM superposition state and show in Fig. 4 the transport fidelities for all 30 MUB states (Methods) conditional on the specific HDBLSM of $|\psi_{00}\rangle_{ab}$. The average transport fidelity is $0.796 \pm 0.066$. Note that all these experimental results are obtained without any background subtraction. The fidelity imperfection mainly arises from two aspects: the prepared states (e.g., entangled states and the to-be transported states) and the imperfect spatio-temporal overlap of photons $a$ and $b$ at BBO2, see Supplementary Note 2 for details.

## Quantum imaging with interaction-free light

Our remote high-dimensional state transport scheme for the above OAM qutrit and ququint states can be straightforwardly extended to embrace any higher-dimensional states. As well known, the vision or image is by far the most universal carrier for transferring information in the real world. According to the digital spiral imaging concept[42,43], any 2D optical image can be equivalently represented by a high-dimensional state vector encoded in the full transverse spatial degrees of freedom. Actually, the aforementioned three- and five-dimensional OAM states can also be seen as the 2D complex images in spatial domain. In this regard, our scheme can also represent another variant of ghost imaging. In the conventional ghost imaging, it necessitates two correlated beams; one beam illuminates an object while the image is recovered from the other beam that has never interacted with the object[44]. However, these two beams actually share a common past, i.e., created from the same pump via SPDC. In contrast, our scheme uses the coherent state $a$ to bear the image information, and then transport nonlocally to photon $c$ after HDBLSM is done. Obviously, the photons used for object illumination and for image recovery do not share any common past, i.e., truly interaction-free. It's noted that there have been several attempts, by using entanglement-swapped photons, to achieve the ghost imaging[45]. Here, specifically, for realizing such a quantum imaging with interaction-free light, Alice and Bob share the photon pairs that are entangled in the full transverse spatial modes (rather than OAM entanglement for perfect vortices), which should be written in terms of LG modes as[31], $|\Psi\rangle_{bc} = \sum_{\ell,p} C_{p,p}^{\ell,-\ell} |\ell,p\rangle_b |-\ell,p\rangle_c$, where $C_{p,p}^{\ell,-\ell}$ denotes the probability amplitude of finding photon $b$ in the standard LG mode of $|\ell,p\rangle$ and the photon $c$ in $|-\ell,p\rangle$, with $\ell$ and $p$ being the azimuthal and radial indices, respectively. Then, at Alice's side, the real-amplitude images, e.g., three-leaf and four-leaf clover, are taken as examples for such a purpose. They are prepared and encoded by coherent beam $a$ with the holograms displayed by SLM1, see the insets of Fig. 1b, c. As mentioned above, the transmission function of any optical image, $O(r,\phi)$, can be equivalently represented by a high-dimensional vector state as, $|\varphi\rangle_a = \sum_{\ell_a,p_a} A_{\ell_a,p_a} |\ell,p\rangle_a$, where $A_{\ell_a,p_a} = \int [\mathrm{LG}_{p_a}^{\ell_a}(r,\phi)]^* O(r,\phi) r dr d\phi$. Similarly, Alice mixes coherent beam $a$ (bearing the image information) and photon $b$ in the BBO2 crystal to conduct SFG, and then filters out the fundamental mode of SFG photons using a SMF that is connected to a PMT. Owing to OAM conservation in the process of SFG[33], the generated SFG photons can be collected by the SMF only when $\ell_a = -\ell_b$ and $p_a = p_b$. Consequently, the photon $c$ at Bob's side is collapsed onto $|\varphi\rangle_c = \sum_{\ell_c,p_c} A_{\ell_c,p_c} \eta_{p_c,p_c}^{\ell_c,-\ell_c} C_{p_c,p_c}^{\ell_c,-\ell_c} |\ell,p\rangle_c$, where $\eta_{p_c,p_c}^{\ell_c,-\ell_c}$ denotes the frequency

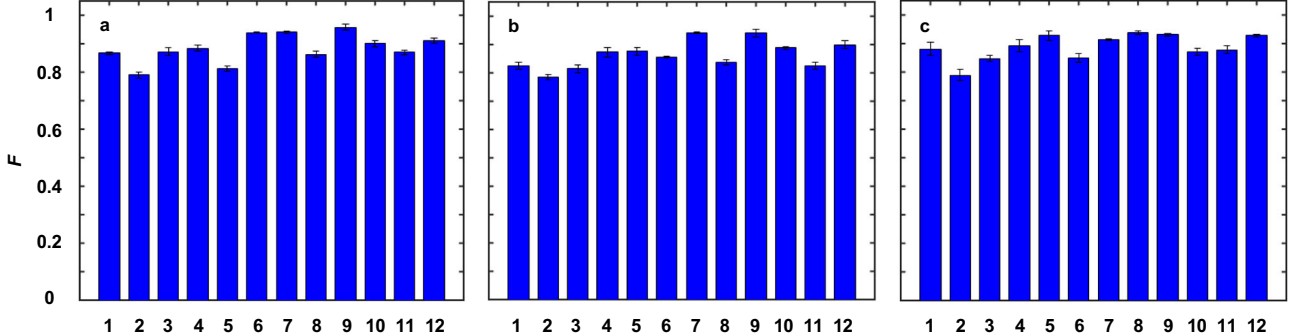

**Fig. 3 | Experimental transport fidelities $F$ for all 12 OAM MUB states conditional on three specific HDBLSM results. a** $|\psi_{00}\rangle_{ab}$, **b** $|\psi_{01}\rangle_{ab}$, and **c** $|\psi_{02}\rangle_{ab}$. The labels of 1-12 in the horizontal axis represent the OAM qutrit states, $|\varphi_1\rangle_a - |\varphi_{12}\rangle_a$. Error bars (standard deviations) are calculated from 3 repetitive independent measurements.

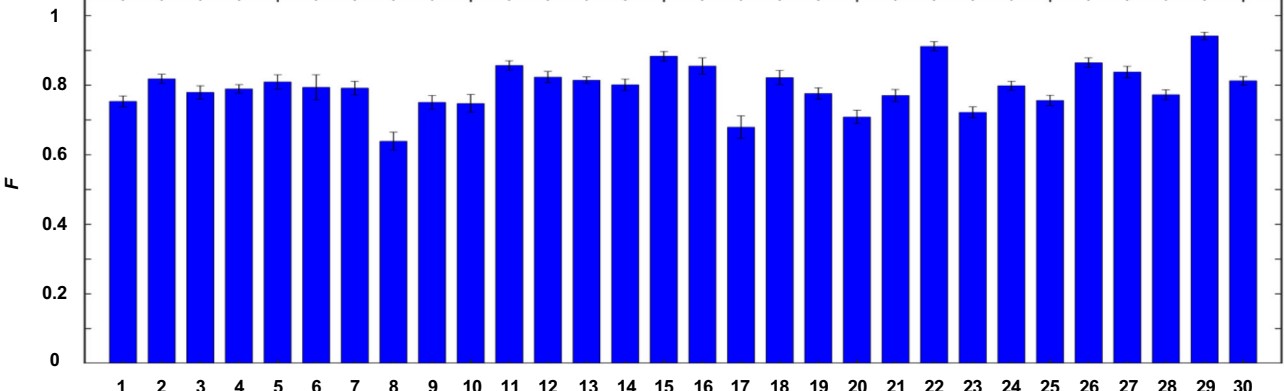

**Fig. 4 | Experimental transport fidelities $F$ for all 30 OAM MUB states conditional on the specific HDBLSM result of $|\psi_{00}\rangle_{ab}$.** The labels of 1-30 in the horizontal axis represent the OAM ququint states, $|\phi_1\rangle_a - |\phi_{30}\rangle_a$ (Methods). Error bars (standard deviations) are calculated from Poissonian counting statistics of the detection events.

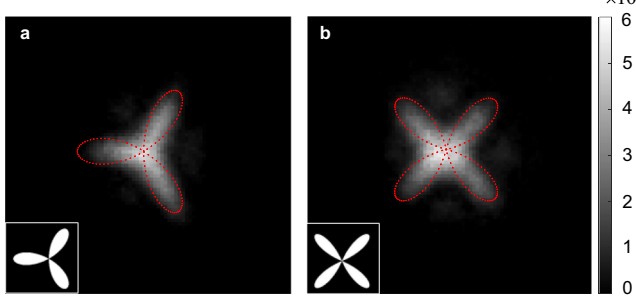

**Fig. 5 | Experimental results for ghost imaging with interaction-free light.**
**a** Three-leaf Clover and **b** four-leaf Clover. Insets show the original images. The gray-scale bar is in units of photons/pixel.

up-conversion efficiency for $LG_{p_{c}}^{\ell_c}$ and $LG_{p_c}^{-\ell_c}$. For sufficiently large and strong pump beams, both $\eta_{p_c, p_c}^{\ell_c, -\ell_c}$ and $C_{p_c, p_c}^{\ell_c, -\ell_c}$ can be approximately a constant, which means that photon $c$ just takes a perfect replica of the original state of coherent beam $a$. Then, the single-photon events from PMT indicates the HDBLSM result $|\psi_{00}\rangle_{ab}$ at Alice's side, which serves the trigger signal for the ICCD camera at Bob's side to capture the image that is transported onto photon $c$.

We present our experimental observations of the interaction-free-light-based ghost imaging of three-leaf and four-leaf Clover images in Fig. 5a, b respectively. As shown by the dotted lines, we define the regions of interest (ROI) for both three-leaf and four-leaf Clover images, which occupy about 432 and 441 pixels, respectively. Here, for a quantitative analysis, we adopt the contrast-to-noise ratio[46], $CNR = (\langle G_{in} \rangle - \langle G_{out} \rangle) / \sqrt{\sigma_{in}^2 + \sigma_{out}^2}$ to characterize the image quality of recorded images, where $\langle G_{in} \rangle$ and $\langle G_{out} \rangle$ are the ensemble average of the photon numbers at any pixel inside and outside the ROI, respectively, while $\sigma_{in}^2$ and $\sigma_{out}^2$ are the corresponding variances. We have CNR = 1.37 and 1.58 for three-leaf and four-leaf Clover images, respectively, which are at the same level with the traditional ghost imaging[47,48], therefore confirming the good performance of our system. Note that, the edges of recorded images in Fig. 5a, b are not sharp compared with the original ones (indicated by insets), which mainly caused by both the non-maximal spatial entanglement for photon pairs generated via SPDC and the nonuniform SFG efficiency for each pixel in the HDBLSM stage. Also, a better time and spatial overlap of all the image planes of the ICCD camera, BBO crystal, and the SLM1 can further improve the CNR.

## Discussion

Here, we use a strong coherent light instead of single photon as the input state to enhance the SFG conversion (~4%) to perform the HDBLSM, and thus realizing the remote transport of high-dimensional OAM states from a coherent beam to a single photon. Therein the knowledge of the coherent source is not used in our scheme, which is similar with one of the key features of teleportation. However, in the original proposal of quantum teleportation, the input unknown state to be teleported should be encoded with single photon. If we adopt the single photon as input source, limited by the rather low frequency conversion efficiency, we need to wait a much longer time for teleportation to occur. In this regard, it is expected that, with the future advances in structured light nonlinear frequency conversion, our scheme can work in the single-photon scenario and will pave the way for realizing truly secure high-dimensional quantum teleportation in the upcoming quantum network.

In our scheme, Alice only needs to send the HDBLSM result via a classical channel while the photon $c$ received by Bob is always in the single-photon state. Then Bob is required to conduct a desired unitary operation on photon $c$, according to the results of HDBLSM

(indicated by Eq. (2) to Eq. (10)), to recover the high-dimensional OAM states correctly. Thus, under ensuring the eavesdropper has no access to the coherent photons $a$, such a procedure can offer an additional security for the information transport and ghost imaging protocol. Whereas, in the direct transport of OAM states, the eavesdropper can acquire the information at any position along the communication channel.

In summary, we have demonstrated a remote transport of high-dimensional OAM superposition states at a distance, by adopting the perfect vortices as the OAM basis for both a successful preparation of high-dimensional maximal entanglement and a better performance of SFG-based HDBLSM. We have experimentally achieved the average transport fidelity of $0.879 \pm 0.048$ and $0.796 \pm 0.066$ for a complete set of 3-dimensional and 5-dimensional MUB, respectively. Further, by exploring the full transverse spatial-mode entanglement, we have succeeded in realizing the ghost imaging of amplitude objects with interaction-free light, which is fundamentally different from previous quantum imaging techniques[49], such as the conventional ghost imaging[44] or quantum imaging with undetected photons[50]. By calculating the CNR of recorded images, we estimate that the imaging performance of this protocol can reach the same level with the traditional ghost imaging. It can be expected that, with the development of the state-of-the-art nonlinear frequency conversion technology, our scheme can be competent to work fully with all single photons in the near future, that is truly high-dimensional quantum teleportation.

## Methods
### Perfect vortices
Photon pairs generated by SPDC have proven to be a reliable entanglement source. However, under the thin-crystal approximation and phase-matching condition with a Gaussian pump beam, the down-converted two-photon OAM entanglement inevitably suffers from the limited spiral bandwidth[31], i.e., lower-order LG modes appears more frequently than higher-order ones. If the standard LG modes are used to represent the OAM eigenstates $|\ell\rangle$, namely, $\langle r, \phi | \ell \rangle = LG_{p=0}^{\ell}(r, \phi)$, the two-photon OAM-entangled state can be written as, $|\Psi\rangle_{bc} = \sum_{\ell} C_{\ell} |\ell\rangle_b | -\ell\rangle_c$, where $C_{\ell, -\ell} = \int LG_0^{\ell}(r, \phi) [LG_0^0(r, \phi)]^* [LG_0^{-\ell}(r, \phi)]^*$ represents the probability amplitude of finding photon $b$ in the mode of $|\ell\rangle$ and the photon $c$ in the mode of $|-\ell\rangle$. Generally, $|\Psi\rangle_{bc}$ is merely a non-maximally entangled OAM state. However, the maximal entanglement is prerequisite for our present scheme. To overcome this obstacle, we adopt the so-called perfect vortices[32] to represent the OAM states, instead of the standard LG modes. In our scheme, we prepare the perfect vortices by modifying the LG modes as[51],

$$MLG_0^{\ell}(r, \phi) = |LG_0^l(r, \phi)| \exp(i\ell\phi) \tag{18}$$

where $l$ is a constant, e.g., $l = 5$. In such a basis of perfect vortices, we know that $C_{\ell, -\ell} = \int LG_0^0(r, \phi) [MLG_0^{\ell}(r, \phi)]^* [MLG_0^{-\ell}(r, \phi)]^*$ will become a constant, as they share the same radial intensity distribution such that the overlap probability is identical. Thus, we can obtain the desired maximally entangled OAM state as, $|\Psi\rangle_{bc} = \frac{1}{\sqrt{d}} \sum_{\ell} |\ell\rangle_b | -\ell\rangle_c$, with $d$ being the dimension of the OAM subspace. Besides, the perfect vortices are also crucial for performing the HDBLSM via SFG. Based on the couple-wave equations describing the SFG, we can estimate the frequency conversion efficiency for our perfect vortices as[24,52,53],

$$\eta_{\ell_a, \ell_b} \propto \int MLG_0^{\ell_a}(r, \varphi) MLG_0^{\ell_b}(r, \varphi) [MLG_0^{\ell_a + \ell_b}(r, \varphi)]^* r dr d\varphi \tag{19}$$

Similarly, we can expect that $\eta_{\ell_a, \ell_b}$ will become a constant even for different OAM modes $\ell_a$ and $\ell_b$, and thus realizing a reliant HDBLSM.

Generally, the computer-generated holographic grating displayed by a spatial light modulator (SLM) is used for OAM generation and

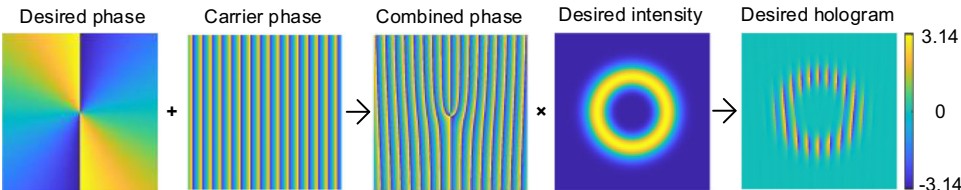

**Fig. 6 | The algorithm of holographic gratings for making perfect vortices.** By modulating the fork grating with a desired intensity, the hologram for preparing the perfect vortex with desired intensity can be realized.

detection experiments. Here, for making perfect vortices, we need to modulate the standard LG modes to let them carry the same radial intensity profiles regardless of different OAM numbers. We illustrate in Fig. 6 the basic principle for making the desired holograms for the perfect vortices. The resultant holograms addressed by SLM can be mathematically described by[54],

$$\Phi(r,\phi)_{\text{SLM}} = (\Phi(r,\phi)_{\text{LG}_0^\ell} + \Phi(r,\phi)_{\text{Linear}})_{\text{mod}2\pi} \times \text{sinc}^2((1 - I(r,\phi)_{\text{LG}_0^\ell})\pi) \tag{20}$$

where $\Phi(r,\phi)_{\text{LG}_0^\ell}$ and $I(r,\phi)_{\text{LG}_0^\ell}$ are desired phase (for different $\ell$) and intensity profiles (e.g., $l = 5$), $\Phi(r,\phi)_{\text{Linear}}$ is the phase of a linear blazed grating, and $\text{sinc}^2(\cdot)$ accounts for the mapping of the phase depth to the diffraction efficiency. From the holographic gratings of Supplementary Fig. 4a–e and the experimental observations of Supplementary Fig. 4f–j, we can see that the different modified LG modes share almost the same intensity profile regardless of carrying different OAM numbers, i.e., the perfect vortices (see Supplementary Note 3 for details).

### Experimental setup

Our experimental setup for the remote transport of high-dimensional OAM states and the ghost imaging with interaction-free light is sketched in Fig. 1. The 140 fs pulsed laser beam with an average power of 1950 mW, a central wavelength of 710 nm and a repetition rate of 80 MHz is focused by lens L1 ($f_1 = 150$mm) and guided to pump a 1.5-mm-long LBO crystal to generate the ultraviolet pulse of a central wavelength of 355 nm and an average power of 200 mW via second-harmonic generation (SHG). The residual 710 nm light beam is collimated by L2 ($f_2 = 100$mm) and guided onto a spatial light modulator (SLM1), which is used to display suitable holographic gratings for preparing coherent beam $a$ in an arbitrary high-dimensional OAM superposition state to be transported. After reflection from SLM1, the light beam has a power of roughly 140 mW. While the generated 355 nm SHG ultraviolet pulses are directed to pump a 3-mm-long β-barium borate crystal (BBO1) to create the frequency non-degeneracy OAM-entangled photon pairs via SPDC, i.e., photon $b$ and $c$, whose wavelengths are centered at 780 nm and 650 nm, respectively. A long-pass dichroic mirror (DM2) is inserted to separate the down-converted photons from the pump. For performing HDBLSM, the 780 nm photon $b$ reflected from DM3 and coherent beam $a$ are combined through DM4, and then sent to 1.5-mm-long BBO2 for implementing SFG. Note that, coherent beam $a$ and photon $b$ are both imaged onto the BBO2 via 4$f$ imaging systems, L7-L9 and L8-L9 ($f_7 = f_8 = 500$mm,$f_9 = 100$mm), to ensure a good spatial overlap of the image planes of BBO2, BBO1 and SLM1. In addition, the temporal overlap between coherent beam $a$ and photon $b$ in BBO2 is realized by accurately adjusting the position of right-angle prism (RAP). Subsequently, we use another 4$f$ system ($f_{10} = 500$mm and $f_{11} = 300$) together with a 4 mm collimated lens to couple the generated 372 nm SFG photon into a single-mode fiber (SMF), which is connected to a photon multiplier tube (PMT). The single-photon event from PMT indicates

the successful HDBLSM signal and herald a transported photon $c$ at Bob's side. For performing the state tomography, photon $c$ is imaged onto SLM2 with a 4$f$ imaging system ($f_3 = 200$mm and $f_4 = 400$mm) and then reimaged onto the input facet of SMF with another 4$f$ imaging system ($f_5 = 500$mm and $f_6 = 10$mm). By switching the flipper mirror (FM), we can capture the transported images by using the ICCD camera, which is triggered by the single-photon events from the PMT. Note that an image preserving optical delay of about 25 m is requested to compensate the electronic delay associated with the PMT and the trigger mechanism in the ICCD camera[47].

### Generalized protocol for remote transport of quidit

Assume Alice wishes to transport to Bob an arbitrary $d$-dimensional OAM state encoded by coherent beam $a$, $|\varphi\rangle_a = \sum_{\ell=\lceil-d/2\rceil}^{\lfloor d/2\rfloor} \alpha_\ell |\ell\rangle_a$, where $\lfloor\cdot\rfloor$ and $\lceil\cdot\rceil$ denote the operation of floor and ceil, respectively, and $\sum|\alpha_\ell|^2 = 1$. Alice and Bob need to share a $d$-dimensional maximally entangled state between photons $b$ and $c$, $|\Psi\rangle_{bc} = \frac{1}{\sqrt{d}}\sum_{\ell=\lceil-d/2\rceil}^{\lfloor d/2\rfloor} |-\ell\rangle_b|\ell\rangle_c$. Then, Alice performs the HDBLSM, $|\psi_{mn}\rangle_{ab} = \frac{1}{\sqrt{d}}\sum_{\ell=\lceil-d/2\rceil}^{\lfloor d/2\rfloor} \exp(i2n\ell\pi/d)|\ell\rangle_a|-\ell\oplus m\rangle_b$, jointly on coherent beam $a$ and photon $b$, where $m,n \in \{0,1,2,\cdots d-1\}$ and $-\ell\oplus m = (-\ell+m)\text{mod }d$. Similarly, by adopting SFG for HDBLSM, we can map all these Bell-like states onto the SFG single-photon states, namely,

$$|\psi_{mn}\rangle_{ab} \rightarrow \frac{1}{\sqrt{d}}\sum_{\ell=\lceil-d/2\rceil}^{\lfloor d/2\rfloor} \exp(i2n\ell\pi/d)|\ell + (-\ell\oplus m)\rangle \tag{21}$$

Note that for $|\psi_{00}\rangle_{ab}$, we get the OAM state of SFG photon as $|\ell = 0\rangle$, which is just orthogonal to those for all the rest Bell-like states and can be directly detected with SMF together with PMT. Turning to our specific case of remote transport of OAM ququint states, conditional on $|\psi_{00}\rangle_{ab}$, we have the photon $c$ in the state, $|\varphi\rangle_c = {}_{ab}\langle\psi_{00}|\Psi\rangle_{abc} = \sum_{\ell=-2}^{2}\alpha_\ell|\ell\rangle$, which is exactly the same with the initial state of coherent beam $a$.

### Mutually unbiased vectors

Generally, the full sets of mutually unbiased bases (MUB) are used to testify the universality of remote transport for all possible super-positions states[40]. Here, for a 5-dimensional system, we need measure 30 states from six mutually unbiased bases (I-VI):

$$\text{I} \rightarrow \begin{cases} |\phi_1\rangle_a = |-2\rangle, \\ |\phi_2\rangle_a = |-1\rangle, \\ |\phi_3\rangle_a = |0\rangle, \\ |\phi_4\rangle_a = |1\rangle, \\ |\phi_5\rangle_a = |2\rangle, \end{cases} \tag{22}$$

$$\text{II} \rightarrow \begin{cases} |\phi_6\rangle_a = |-2\rangle + |-1\rangle + |0\rangle + |1\rangle + |2\rangle, \\ |\phi_7\rangle_a = |-2\rangle + \gamma|-1\rangle + \gamma^2|0\rangle + \gamma^3|1\rangle + \gamma^4|2\rangle, \\ |\phi_8\rangle_a = |-2\rangle + \gamma^2|-1\rangle + \gamma^4|0\rangle + \gamma|1\rangle + \gamma^3|2\rangle, \\ |\phi_9\rangle_a = |-2\rangle + \gamma^3|-1\rangle + \gamma|0\rangle + \gamma^4|1\rangle + \gamma^2|2\rangle, \\ |\phi_{10}\rangle_a = |-2\rangle + \gamma^4|-1\rangle + \gamma^3|0\rangle + \gamma^2|1\rangle + \gamma|2\rangle, \end{cases} \quad (23)$$

$$\text{III} \rightarrow \begin{cases} |\phi_{11}\rangle_a = |-2\rangle + \gamma|-1\rangle + \gamma^4|0\rangle + \gamma^4|1\rangle + \gamma|2\rangle, \\ |\phi_{12}\rangle_a = |-2\rangle + \gamma^2|-1\rangle + \gamma|0\rangle + \gamma^2|1\rangle + |2\rangle, \\ |\phi_{13}\rangle_a = |-2\rangle + \gamma^3|-1\rangle + \gamma^3|0\rangle + |1\rangle + \gamma^4|2\rangle, \\ |\phi_{14}\rangle_a = |-2\rangle + \gamma^4|-1\rangle + |0\rangle + \gamma^3|1\rangle + \gamma^3|2\rangle, \\ |\phi_{15}\rangle_a = |-2\rangle + |-1\rangle + \gamma^2|0\rangle + \gamma|1\rangle + \gamma^2|2\rangle, \end{cases} \quad (24)$$

$$\text{IV} \rightarrow \begin{cases} |\phi_{16}\rangle_a = |-2\rangle + \gamma^2|-1\rangle + \gamma^3|0\rangle + \gamma^3|1\rangle + \gamma^2|2\rangle, \\ |\phi_{17}\rangle_a = |-2\rangle + \gamma^3|-1\rangle + |0\rangle + \gamma|1\rangle + \gamma|2\rangle, \\ |\phi_{18}\rangle_a = |-2\rangle + \gamma^4|-1\rangle + \gamma^2|0\rangle + \gamma^4|1\rangle + |2\rangle, \\ |\phi_{19}\rangle_a = |-2\rangle + |-1\rangle + \gamma^4|0\rangle + \gamma^2|1\rangle + \gamma^4|2\rangle, \\ |\phi_{20}\rangle_a = |-2\rangle + \gamma|-1\rangle + \gamma|0\rangle + |1\rangle + \gamma^3|2\rangle, \end{cases} \quad (25)$$

$$\text{V} \rightarrow \begin{cases} |\phi_{21}\rangle_a = |-2\rangle + \gamma^3|-1\rangle + \gamma^2|0\rangle + \gamma^2|1\rangle + \gamma^3|2\rangle, \\ |\phi_{22}\rangle_a = |-2\rangle + \gamma^4|-1\rangle + \gamma^4|0\rangle + |1\rangle + \gamma^2|2\rangle, \\ |\phi_{23}\rangle_a = |-2\rangle + |-1\rangle + \gamma|0\rangle + \gamma^3|1\rangle + \gamma|2\rangle, \\ |\phi_{24}\rangle_a = |-2\rangle + \gamma|-1\rangle + \gamma^3|0\rangle + \gamma|1\rangle + |2\rangle, \\ |\phi_{25}\rangle_a = |-2\rangle + \gamma^2|-1\rangle + |0\rangle + \gamma^4|1\rangle + \gamma^4|2\rangle, \end{cases} \quad (26)$$

$$\text{VI} \rightarrow \begin{cases} |\phi_{26}\rangle_a = |-2\rangle + \gamma^4|-1\rangle + \gamma|0\rangle + \gamma|1\rangle + \gamma^4|2\rangle, \\ |\phi_{27}\rangle_a = |-2\rangle + |-1\rangle + \gamma^3|0\rangle + \gamma^4|1\rangle + \gamma^3|2\rangle, \\ |\phi_{28}\rangle_a = |-2\rangle + \gamma|-1\rangle + |0\rangle + \gamma^2|1\rangle + \gamma^2|2\rangle, \\ |\phi_{29}\rangle_a = |-2\rangle + \gamma^2|-1\rangle + \gamma^2|0\rangle + |1\rangle + \gamma|2\rangle, \\ |\phi_{30}\rangle_a = |-2\rangle + \gamma^3|-1\rangle + \gamma^4|0\rangle + \gamma^3|1\rangle + |2\rangle, \end{cases} \quad (27)$$

where $\gamma = \exp(i2\pi/5)$.

## Data availability
The datasets generated and analysed during this study are available from the corresponding author upon request. Source data for Figs. 2–4 are provided with this paper. Source data are provided with this paper.

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

## Acknowledgements

This work was supported by the National Natural Science Foundation of China (12034016, 61975169), the National Key R&D Program of China (2023YFA1407200), China Postdoctoral Science Foundation (2021M691891), the Natural Science Foundation of Fujian Province of China (2021J02002), and the program for New Century Excellent Talents in University of China (NCET-13-0495).

## Author contributions

L.C. conceived the idea. L.C. and X.Q. designed the experiment. X.Q. conducted the experiment with the help from H.G and L.C. X.Q. and L.C. analyzed the data and co-wrote the manuscript. L.C. supervised the project.

## Competing interests

The authors declare no competing interests.
