## [Peer Review File · Nature Communications]

Remote transport of high-dimensional orbital angular momentum states and ghost images via spatial-mode-engineered frequency conversionREVIEWER COMMENTS

Reviewer #1 (Remarks to the Author):

The authors present a paper titled “Teleportation-based quantum imaging,” and this work is of significant interest but does not come without significant controversy.

I do not believe that what the authors present here is the teleportation of a quantum state. For that to happen, the state that is being teleported has to belong to a single-photon state, but in this case presented in this work, the state belongs to a “strong coherent wave packet”. As such, there are many photons in this state, and only one of them is destroyed and teleported to the distant system. The system presented here is some form of a hybrid classical and quantum system – it is not exclusively a quantum system. The way that the authors present the work is, however, in the language of quantum, and therefore they open themselves up for criticism. For example, the authors discuss the high-dimensional entanglement and reconstruct high-dimensional density matrices, for example, “5D OAM entangled states” (supplementary information). However, if I am correct, this is the shared state between Alice’s “strong coherent wave packet” and Bob’s “single photon”. This is not the density matrix associated with the state of two single photons. It is a hybrid quantum/classical state, where one of the parties, Alice, has a system that can be described classically, and Bob’s system is a single quantum system. I would criticise this even further by stating that it is a fundamental requirement of traditional quantum entanglement that any outcome of a measurement on either Alice’s or Bob’s system has uncertainty; it is the combined measurement that has certainty. The experiment presented here has no uncertainty for Alice’s system. The states are created by taking a strong laser field that is incident on an SLM. There is no uncertainty in this measurement – it is the generation of a strong coherent field in a particular mode of light. This field is then combined at BB02 with a single photon that was generated at BBO1. This can generate blue light at BBO2, and it is this process that the authors use to teleport the state. But I note again, there is no uncertainty in the state of Alice, and this is a fundamental aspect of entanglement, and I believe a fundamental aspect of quantum teleportation.

On a more positive note, I do not want to question the heroic effort that is required to

execute an experiment of this nature. There is a significant amount of work that is required to create and execute this experiment. However, the interpretation of the data is not what I would have presented it as it is. I believe that this is original research that is incredibly interesting. I note also that a similar experiment has been reported on the Arxiv (<https://arxiv.org/abs/2111.13624>). A comment on this experiment included in the publication, "On single-photon can classical interference" (<https://iopscience.iop.org/article/10.1088/1402-4896/ac971a>). I will paste the relevant extract of that paper here:

One important example is a recent realisation of quantum teleportation in which the state to be teleported is encoded first on a laser field and is then teleported to a distant single photon. One nonlinear optical process produces a pair of entangled photons as is often the starting point in a teleportation experiment. The comparison step is achieved by a frequency conversion arrangement in which one of the entangled photons (the local photon) interacts with the laser field to produce a new photon carrying information from both the previously entangled photon and the laser field. The teleportation may be viewed as the transfer of the state of one of the laser photons to the distant and previously entangled photon. As with single-photon interference, this laser-based teleportation does not share all of the features of a single-photon teleportation; it cannot teleport entanglement for example. It does, nevertheless, transfer the state of the laser photons to the distant photon and it does this without using knowledge of the state of the laser photons and this, of course, is the key feature of quantum teleportation.

As you can see, there are aspects of the work here and the other paper that are common to quantum teleportation. However, it concerns me that the authors of the current paper make claims regarding high-dimensional entanglement, and this is not allowed according to my current understanding and confirmed by the reference above.

It is for this reason that I cannot recommend publication in Nature Communications without a significant re-write of the claims of the current version. I recommend that authors consider re-phrasing the work without the need to refer to traditional quantum teleportation. The work here is new, it is novel, but it is not quantum teleportation in the

traditional sense. I also note that the authors claim that they are, “realizing the first teleportation-based quantum image transport at a distance”. However, I would say that the work presented in “Ghost imaging using entanglement-swapped photons” (<https://www.nature.com/articles/s41534-019-0176-5>) teleports an image, even if the final quantum state is mixed.

Reviewer #2 (Remarks to the Author):

The manuscript entitled “Teleportation-based quantum imaging” presents a proof-of-concept demonstration of a non-linear scheme that allows for the transfer, without any physical link, of high-dimensional orbital angular momentum (OAM) states between two distant parties. The communication is realised through the projective measurement of an arbitrary OAM state using a pair of high-dimensional entangled photons, and a joint measurement mediated through sum-frequency generation (SFG) with a strong coherent state. The protocol is demonstrated with projections over the complete set of MUB states in three and five dimensions. Furthermore, they use the multi-modal capabilities of the scheme to transfer transverse spatial information encoded in optical images.

This up-conversion-aided technique was initially proposed as a quantum teleportation strategy that can overcome fundamental limitations in linear optical teleportation [Opt. Comm. 193 175 (2001), Phys. Rev. A 76, 033801 (2007)]. However, as is pointed out by the authors, real teleportation would require a single photon as the input, which ensures the security of the transfer and the capability of transporting entanglement. Consequently, the authors refer to their work as a ‘teleportation-based’ technique that uses a bright coherent state for improving the efficiency of the up-conversion process. The method allows for arbitrary, high-dimensional state projections, resembling measurement strategies in the time-frequency domain like the Quantum Pulse Gate [Opt. Express 19, 13770 (2011), Phys. Rev. A 90, 030302 (2014)], or the coherent measurement of time-bin encoded photons [PRL 111, 153602 (2013)].

The results are presented along with an explanation of how a joint measurement of the coherent state and one of the entangled photons is made through the SFG process. This constitutes the high-dimensional Bell-like state measurement that is key for achieving the transfer. To ensure the unbiasedness of the projection, the “perfect vortex” technique is

employed as an alternative to procrustean filtering, which is commonly used in spatial mode measurements. Complete tomographical measurements in 3 and 5 dimensions demonstrate the experimental capabilities of the implementation, complemented by an analysis of the sources of errors that affect the fidelities of the transferred states. The authors compare these fidelities to a classical bound [Ref. [43], Phys. Rev. Lett. 72, 797 (1994), and Phys. Rev. Lett. 74, 1259 (1995))] that considers only one copy of the state to be transferred. However, this comparison is inadequate because the use of a coherent state introduces the possibility of having multiple copies. In that sense, the advantage when compared to the case of a completely classical channel between Alice and Bob is not clear and would require further discussion.

Encoding the input information into a coherent state enhances the efficiency of the technique but compromises the security of the communication; furthermore, it removes its advantage when compared to the use of a classical channel. The absence of these characteristics, which are essential in the context of quantum technologies, should prevent the authors to refer to their experiment as "teleportation-based" or as a "slight" deviation from quantum teleportation. This is an inaccurate comparison to high-dimensional quantum teleportation and the authors should reformulate their claims and change their title, clarifying that their work serves only as a study of the high-dimensional capabilities of the non-linear projection, which could become relevant when an increase in the efficiency of SHG processes allows for implementations of quantum state teleportation through practical non-linear high-dimensional Bell-state measurements.

In addition, significant changes need to be made regarding the use of the technique for the transfer of optical images. Conceptually, introducing optical image transport as an implementation related to quantum imaging or teleportation-based is misleading. The use of these terms indicates the coherent transfer of information. Contrary to the proposal for quantum teleportation of a multimode field [Opt. Comm. 193 175 (2001)], the transferred information in the present experiment is not a spatially encoded quantum state, but the intensity information of an optical image encoded in a transverse spatial basis. The use of a non-standard measure of "image fidelity", which relies only on intensity measurements of the prepared and measured image, neglects all phase information and cannot be compared to any quantum state fidelity or any non-classical bound. Furthermore, deductions of the capacity of a quantum channel (i.e., a channel that supports the coherent transport of

quantum information) from these measurements make no sense because of the lack of evidence for coherence. The results only support the transfer of the amplitude of the encoded image, and thus, any claim of teleportation-based quantum transport is invalid and needs to be removed from the manuscript.

This manuscript can be of significance to the field because it demonstrates a technique for arbitrary projections of OAM states in up to five dimensions, allowing for the coherent transfer of spatial information. Nevertheless, I can only recommend the publication if the previous concerns are addressed, and the relevant changes are implemented.

Response to Reviewer #1

Comment 1. The authors present a paper titled “Teleportation-based quantum imaging,” and this work is of significant interest but does not come without significant controversy.

Response. Thank you very much for the positive recommendations and the instructive suggestions. Following your suggestions, we have tried our best to rephrase our work therein tightly focusing on what we have really done, i.e., remote transport of high-dimensional OAM states from a coherent light beam to a single photon, and without referring to the traditional quantum teleportation. By addressing your concerns, we indeed feel that our manuscript has been much improved now and can effectively avoid the controversy, and we sincerely wish you could consider it in Nature communications.

Comment 2. I do not believe that what the authors present here is the teleportation of a quantum state. For that to happen, the state that is being teleported has to belong to a single-photon state, but in this case presented in this work, the state belongs to a “strong coherent wave packet”. As such, there are many photons in this state, and only one of them is destroyed and teleported to the distant system. The system presented here is some form of a hybrid classical and quantum system – it is not exclusively a quantum system. The way that the authors present the work is, however, in the language of quantum, and therefore they open themselves up for criticism. For example, the authors discuss the high-dimensional entanglement and reconstruct high-dimensional density matrices, for example, “5D OAM entangled states” (supplementary information). However, if I am correct, this is the shared state between Alice’s “strong coherent wave packet” and Bob’s “single photon”. This is not the density matrix associated with the state of two single photons. It is a hybrid quantum/classical state, where one of the parties, Alice, has a system that can be described classically, and Bob’s system is a single quantum system. I would criticise this even further by stating that it is a fundamental requirement of traditional quantum entanglement that any outcome of a measurement on either Alice’s or Bob’s system has uncertainty; it is the combined measurement that has certainty. The experiment presented here has no uncertainty for

Alice's system. The states are created by taking a strong laser field that is incident on an SLM. There is no uncertainty in this measurement – it is the generation of a strong coherent field in a particular mode of light. This field is then combined at BB02 with a single photon that was generated at BBO1. This can generate blue light at BBO2, and it is this process that the authors use to teleport the state. But I note again, there is no uncertainty in the state of Alice, and this is a fundamental aspect of entanglement, and I believe a fundamental aspect of quantum teleportation.

Response. It's correct for the reviewer to point out that *“For that to happen, the state that is being teleported has to belong to a single-photon state, but in this case presented in this work, the state belongs to a “strong coherent wave packet””*. In our scheme, we have used a strong coherent state as the input to enhance the SFG efficiency, and thus realizing the remote transport of high-dimensional OAM states from a coherent beam to a single photon without direct transmission of information carriers. Indeed, the input state here contains many photons, which means that there are many copies of the unknown state to be teleported, which deviates from the original proposal of quantum teleportation. Therefore, we agree well with the reviewer that it is not appropriate to say that our present experiment is the teleportation of a quantum state and that will cause controversy. For clarifying this point, we have focused only on what we did and rewritten our manuscript. Specifically, we have removed the description of quantum teleportation from the sections of abstract, introduction, theory, and experimental results. And merely in the section of discussion, we conduct an outlook that our scheme could be extend to the single photon scenario and pave the way for realizing the high-dimensional quantum teleportation with the future advances in structured light nonlinear frequency conversion (see also our **Response to Comment 3**).

As for “high-dimensional entanglement states”, we would like to clarify our protocol here. For realizing the transport of high-dimensional OAM states without direct transmission of information carriers (taking 3-dimensional state for example), Alice and Bob need share the high-dimensional and maximally entangled state (photons

b and c), $|\Psi\rangle_{bc} = \frac{1}{\sqrt{3}}(|-1,1\rangle_{bc} + |0,0\rangle_{bc} + |1,-1\rangle_{bc})$, while high-dimensional OAM states to be transported are carried by the coherent beam a . Then, Alice mixes the coherent light a and photon b in the nonlinear crystal to conduct the sum-frequency generation (SFG). Owing to the OAM conservation in SFG³³, if we detect the SFG photon with a single mode fiber (SMF), which can only support fundamental mode, thus then, the topological charges of coherent light a and photon b satisfy the relation, $\ell_a + \ell_b = 0$. Also, owing to the OAM conservation in SPDC, the coincidence between SFG photon and photon c can only be detected when $\ell_a = -\ell_b = \ell_c$. Accordingly, the OAM state of the coherent light a could be transported to photon c without direct transfer the state itself. Here, the high-dimensional entanglement between Alice and Bob can be seen as the information channel. And in order to realize the perfect transport, therein a maximally entanglement state is required. Thus, we adopt “perfect vortex” to prepare maximally OAM entanglement, and reconstruct the density matrices to quantitatively characterize. Noted that the photons b and c constitute an entangled quantum system, while the coherent beam a is a classical system. And the aforementioned mixing the coherent light a and photon b in the nonlinear crystal to conduct the sum-frequency generation (SFG) at Alice side could be seen as a Bell-like state measurement, which connects the classical system (coherent light a) and quantum system (photon b). Accordingly, it’s correct for the reviewer to point out that “*the system presented here is some form of a hybrid classical and quantum system*”.

In our revised manuscript, we have added some remarks to clarify this point (See Lines 86-114 in pages 5-7)

“...Without loss of generality, we first consider the transport of 3-dimensional OAM states. Assume Alice wishes to transport to Bob the OAM state, $|\varphi\rangle_a = \alpha_1|-1\rangle_a + \alpha_2|0\rangle_a + \alpha_3|1\rangle_a$, which is bore by a strong coherent light a , and the complex coefficients $\alpha_1, \alpha_2, \alpha_3$ fulfill $|\alpha_1|^2 + |\alpha_2|^2 + |\alpha_3|^2 = 1$. For transporting such an arbitrary 3-dimensional states, Alice and Bob need share the aforementioned high-

dimensional and maximally entangled state (photons b and c),

$$|\Psi\rangle_{bc} = \frac{1}{\sqrt{3}}(|-1,1\rangle_{bc} + |0,0\rangle_{bc} + |1,-1\rangle_{bc}). \quad (1)$$

Then, Alice mixes the coherent light a and photon b in the nonlinear crystal to conduct the sum-frequency generation (SFG). Owing to the OAM conservation in SFG³³, if we detect the SFG photon with a single mode fiber (SMF), which can only support fundamental mode, thus then, the topological charges of coherent light a and photon b satisfy the relation, $\ell_a + \ell_b = 0$. Also, owing to the OAM conservation in SPDC, seen in Eq. (1), the coincidence between SFG photon and photon c can only be detected when $\ell_a = -\ell_b = \ell_c$. Accordingly, the OAM state of the coherent light a could be transported to photon c without direct transfer the state itself. However, due to the non-uniform frequency conversion efficiency for different input OAM eigenstates, e.g., the LG modes, the transport performance will get worse as the increase of OAM value. Here, we adopt the perfect vortices to overcome this obstacle, as they can guarantee the identical conversion efficiency for different pairwise OAM modes $|\ell\rangle_a |-\ell\rangle_b$. Thus, the high-dimensional OAM states can be transported reliably (Methods).

In other words, after nonlinear projection via SMF, the OAM states of coherent light a and converted photon b satisfy $|\psi\rangle_{ab} = \frac{1}{\sqrt{3}}(|-1,1\rangle_{ab} + |0,0\rangle_{ab} + |1,-1\rangle_{ab})$. Noted that, due to the coherent light a adopted here, $|\psi\rangle_{ab}$ is just used to characterize the dependence relationship between a and b , and does not represent a regular entanglement state. In other words, it can be seen as a hybrid classical and quantum system. Interestingly, the relationship $|\psi\rangle_{ab}$ is formally similar with one of the 3-dimensional Bell states, $|\psi_{mn}\rangle = \frac{1}{\sqrt{3}} \sum_{\ell=-1}^1 \exp(i 2n\ell\pi / 3) |\ell\rangle |-\ell \oplus m\rangle$ with $m, n \in \{0,1,2\}$. Thus here, we termed it as the high-dimensional Bell-like measurement...”

Comment 3. On a more positive note, I do not want to question the heroic effort that is required to execute an experiment of this nature. There is a significant amount of work

that is required to create and execute this experiment. However, the interpretation of the data is not what I would have presented it as it is. I believe that this is original research that is incredibly interesting. I note also that a similar experiment has been reported on the Arxiv (<https://arxiv.org/abs/2111.13624>). A comment on this experiment included in the publication, “On single-photon can classical interference” (<https://iopscience.iop.org/article/10.1088/1402-4896/ac971a>). I will paste the relevant extract of that paper here:

One important example is a recent realisation of quantum teleportation in which the state to be teleported is encoded first on a laser field and is then teleported to a distant single photon. One nonlinear optical process produces a pair of entangled photons as is often the starting point in a teleportation experiment. The comparison step is achieved by a frequency conversion arrangement in which one of the entangled photons (the local photon) interacts with the laser field to produce a new photon carrying information from both the previously entangled photon and the laser field. The teleportation may be viewed as the transfer of the state of one of the laser photons to the distant and previously entangled photon. As with single-photon interference, this laser-based teleportation does not share all of the features of a single-photon teleportation; it cannot teleport entanglement for example. It does, nevertheless, transfer the state of the laser photons to the distant photon and it does this without using knowledge of the state of the laser photons and this, of course, is the key feature of quantum teleportation.

As you can see, there are aspects of the work here and the other paper that are common to quantum teleportation. However, it concerns me that the authors of the current paper make claims regarding high-dimensional entanglement, and this is not allowed according to my current understanding and confirmed by the reference above. It is for this reason that I cannot recommend publication in Nature Communications without a significant re-write of the claims of the current version. I recommend that authors consider re-phrasing the work without the need to refer to traditional quantum teleportation. The work here is new, it is novel, but it is not quantum teleportation in the traditional sense. I also note that the authors claim that they are, “realizing the first

teleportation-based quantum image transport at a distance”. However, I would say that the work presented in “Ghost imaging using entanglement-swapped photons” (<https://www.nature.com/articles/s41534-019-0176-5>) teleports an image, even if the final quantum state is mixed.

Response. Thank you very much for recognizing the “original” and “incredibly interesting” of our work. As you indicate, there are some features of the present work that are common to quantum teleportation. That is remotely transporting high-dimensional OAM states of the coherent beam to the distant and previously entangled photon without using knowledge of the state of the laser photons, which can also be seen in the aforementioned publication, “On single-photon and classical interference” (<https://iopscience.iop.org/article/10.1088/1402-4896/ac971a>). Also, our present scheme cannot work under the signal photon scenario, thus cannot teleport entanglement, which deviates from the original proposal of quantum teleportation. Accordingly, terming it as teleportation or teleportation-based will cause significant controversy. Furthermore, thank you for introducing the similar experiment work (<https://arxiv.org/abs/2111.13624>). Actually, both sides are submitting works back-to-back. So, we will consult with the editor about how to properly cite it according to the requirements of the journal.

Thank you very much again for these insightful and very inspiring comments. By addressing your concerns, we indeed feel that our work has been much improved. We have tried our best to consider your suggestions and rephrase the manuscript accordingly, which has clearly reflected the two levels of our scheme. On one hand, it provides a remote way to transport high-dimensional OAM states without direct transmission of information carries. On the other hand, it represents a ghost imaging with truly interaction-free light when working with a strong coherent state. Accordingly, we have made the following changes.

In the section of Title and Abstract.

We have rewritten the title and abstract to reflect what we have really done (See Lines 1-24 in Pages 1-2)

“Title: Remote transport of high-dimensional orbital angular momentum states:
Towards quantum imaging with interaction-free light

Abstract: The efficient transport and engineering photonic orbital angular momentum (OAM) lie at the heart of various related classical and quantum applications. Here, by leveraging the spatial-mode-engineered frequency conversion, we realize the remote transport of high-dimensional orbital angular momentum (OAM) states between two distant parties without direct transmission of information carriers. We exploit “perfect vortices” for preparing high-dimensional yet maximal OAM entanglement. Based on nonlinear sum-frequency generation working with a strong coherent wave packet and a single photon, we conduct the Bell-like state measurements for high-dimensional “perfect vortices”. We experimentally achieve the average transport fidelity 0.879 ± 0.048 and 0.796 ± 0.066 for a complete set of 3-dimensional and 5-dimensional OAM mutually unbiased bases, respectively. Furthermore, by exploring the full transverse entanglement, we construct another strategy of quantum imaging with interaction-free light. It is expected that, with the future advances in nonlinear frequency conversion, our scheme will pave the way for realizing truly secure high-dimensional quantum teleportation in the upcoming quantum network.”

In the section of Introduction.

We have removed connections with teleportation and merely focused on the transport of OAM and nonlinear frequency conversion (See Lines 25-61 in Pages 3-4)

“Since the seminal discovery by Allen *et al.*¹, orbital angular momentum (OAM) of light has aroused numerous attentions as its high-dimensional nature provides the attractive potentials in some classical and quantum information protocols²⁻⁵, e.g., realizing large-alphabet optical communication^{6,7} and constructing the high-dimensional photonic quantum entanglement⁸, as well as provides the toolkit for the investigation of quantum fundamental questions⁹. However, these benefits are only conceivable when we are able to efficiently generate, manipulate, transport, and detect the OAM state. Hitherto, extensive investigations of OAM transport^{6,7,10-17} have been

reported, such as OAM transport in free-space, specialized optical fiber, and even the distribution of OAM entanglement. Whereas, it is noted that the most of these previous schemes focused on the direct information transport only.

Recent years have also witnessed a growing interest in the nonlinear frequency conversion of the structured light^{5,18}. In spatial domain, it was demonstrated that using OAM light as fundamental frequency light source, one can realize the image processing and mode detection during the process of upconversion imaging^{19,20}, and by shaping the spatial shape of the pump source, one can enhance the performance of high-dimensional OAM frequency interface²¹⁻²⁴. In time-frequency domain, by using the dispersion-engineered sum-frequency generation, the quantum pulse gate for harnessing and selecting the time-frequency mode has been realized^{25,26}. Also, it has provided a powerful tool for time-bin entanglement measurement²⁷. Particularly, we note that there have several attempts, by means of sum-frequency generation, to perform the coherent transfer of the polarization state between two distant parties without direct transmission of the information²⁸, and to realize the entanglement swapping of time-bin entanglement^{29,30}, however, the scenario of high-dimensional OAM state has not yet been fully explored. Some interesting questions arise naturally: whether the nonlinear frequency conversion can be used to transport the high-dimensional OAM state with an unprecedented way, and what new features this might lead to. These also form the major incentive of our present work.

Here, by leveraging the spatial-mode-engineered frequency conversion, we realize the remote transport of high-dimensional orbital angular momentum (OAM) states at a distance without direct transmission of information carriers. Specifically, by exploiting perfect vortices, we prepare high-dimensional yet maximally entangled orbital angular momentum (OAM) states as communication channel. And then, we employ sum-frequency generation (SFG) with a strong coherent state, which is used to bear the high-dimensional OAM state, to enhance the frequency conversion, and thus performing the perfect-vortex-based high-dimensional Bell-like state measurement reliably. Furthermore, we report the quantum imaging with truly interaction-free light.”

In the section of ghost imaging with interaction-free light.

Thank you for introducing this relevant reference [NPJ Quantum Information 5, 63 (2019)], we are happy to let you know that, as inspired by both your suggestion and also the comments by Reviewer #2, we clearly reveal the connection between our scheme and a new variant of ghost imaging protocol, that is ghost imaging with interaction-free light. We have clarified this point in our revision (See Lines 228-236 in Pages 13-14)

“... In the conventional ghost imaging, it necessitates two correlated beams; one beam illuminates an object while the image is recovered from the other beam that has never interacted with the object⁴⁴. However, these two beams actually share a common past, i.e., created from the same pump via SPDC. In contrast, our scheme uses the coherent state a to bear the image information, and then transport nonlocally to photon c after HDBLSM is done. Obviously, the photons used for object illumination and for image recovery do not share any common past, i.e., truly interaction-free. It's noted that there have also several attempts, by using entanglement-swapped photons, to achieve the ghost imaging⁴⁵...”

In the section of Discussion.

We conduct an outlook that our scheme could be extend to the single photon scenario and pave the way for realizing the high-dimensional quantum teleportation with the future advances in structured light nonlinear frequency conversion (See Lines 283-302 in Pages 16-17)

“Here, we use a strong coherent light instead of single photons as the input state to enhance the SFG conversion ($\sim 4\%$) to perform the HDBLSM, and thus realizing the remote transport of high-dimensional OAM states from a coherent beam to a single photon. Therein the knowledge of the coherent source is not used in our scheme, which is similar with the one of key features of teleportation. However, in the original proposal of quantum teleportation, the input unknown state to be teleported should be encoded with single photon. If we adopt the single photon as input source, limited by the rather low frequency conversion efficiency, we need to wait a much longer time for teleportation to occur. In this regard, it is expected that, with the future advances in

structured light nonlinear frequency conversion, our scheme can work in the single photon scenario and will pave the way for realizing truly secure high-dimensional quantum teleportation in the upcoming quantum network.

In our scheme, Alice only needs to send the HDBLSM result via a classical link while the photon c received by Bob is always in the single-photon state. Then Bob is required to conduct a desired unitary operation on photon c , according to the results of HDBLSM (indicated by Eq. (2)), to recover the high-dimensional OAM states correctly. Thus, under ensuring the eavesdropper has no access to the coherent photons a , such a procedure can offer an additional security for the information transport and ghost imaging protocol. Whereas, in the direct transport of OAM states, the eavesdropper can acquire the information at any position along the communication channel.”

Response to Reviewer #2

Comment 1. The manuscript entitled “Teleportation-based quantum imaging” presents a proof-of-concept demonstration of a non-linear scheme that allows for the transfer, without any physical link, of high-dimensional orbital angular momentum (OAM) states between two distant parties. The communication is realised through the projective measurement of an arbitrary OAM state using a pair of high-dimensional entangled photons, and a joint measurement mediated through sum-frequency generation (SFG) with a strong coherent state. The protocol is demonstrated with projections over the complete set of MUB states in three and five dimensions. Furthermore, they use the multi-modal capabilities of the scheme to transfer transverse spatial information encoded in optical images.

This up-conversion-aided technique was initially proposed as a quantum teleportation strategy that can overcome fundamental limitations in linear optical teleportation [Opt. Comm. 193 175 (2001), Phys. Rev. A 76, 033801 (2007)]. However, as is pointed out by the authors, real teleportation would require a single photon as the input, which ensures the security of the transfer and the capability of transporting entanglement. Consequently, the authors refer to their work as a ‘teleportation-based’ technique that uses a bright coherent state for improving the efficiency of the up-conversion process. The method allows for arbitrary, high-dimensional state projections, resembling measurement strategies in the time-frequency domain like the Quantum Pulse Gate [Opt. Express 19, 13770 (2011), Phys. Rev. A 90, 030302 (2014)], or the coherent measurement of time-bin encoded photons [PRL 111, 153602 (2013)].

Comment 2. The results are presented along with an explanation of how a joint measurement of the coherent state and one of the entangled photons is made through the SFG process. This constitutes the high-dimensional Bell-like state measurement that is key for achieving the transfer. To ensure the unbiasedness of the projection, the “perfect vortex” technique is employed as an alternative to procrustean filtering, which is commonly used in spatial mode measurements. Complete tomographical measurements in 3 and 5 dimensions demonstrate the experimental capabilities of the

implementation, complemented by an analysis of the sources of errors that affect the fidelities of the transferred states. The authors compare these fidelities to a classical bound [Ref. [43], Phys. Rev. Lett. 72, 797 (1994), and Phys. Rev. Lett. 74, 1259 (1995)] that considers only one copy of the state to be transferred. However, this comparison is inadequate because the use of a coherent state introduces the possibility of having multiple copies. In that sense, the advantage when compared to the case of a completely classical channel between Alice and Bob is not clear and would require further discussion.

Comment 3. Encoding the input information into a coherent state enhances the efficiency of the technique but compromises the security of the communication; furthermore, it removes its advantage when compared to the use of a classical channel. The absence of these characteristics, which are essential in the context of quantum technologies, should prevent the authors to refer to their experiment as "teleportation-based" or as a "slight" deviation from quantum teleportation. This is an inaccurate comparison to high-dimensional quantum teleportation and the authors should reformulate their claims and change their title, clarifying that their work serves only as a study of the high-dimensional capabilities of the non-linear projection, which could become relevant when an increase in the efficiency of SHG processes allows for implementations of quantum state teleportation through practical non-linear high-dimensional Bell-state measurements.

Response. We would like to answer the above comments together. It is very instructive for the reviewer to point out these questions. We agree well with the reviewer that "*real teleportation would require a single photon as the input, which ensures the security of the transfer and the capability of transporting entanglement.*" And the coherent light adopted in our present scheme indeed deviates from the original proposal of quantum teleportation, thus claiming our scheme as "teleportation-based" is inappropriate and will cause controversy. Following the reviewer's suggestions, we have removed the connection with teleportation and changed title as "**Remote transport of high-dimensional orbital angular momentum states: Towards quantum imaging with**

interaction-free light", which exactly represents what we have really done at current stage. And merely at the end of the manuscript, we discuss that our scheme could be extended to single photon scenario with the future advances in nonlinear frequency conversion. Correspondingly, we have also removed the connections with teleportation in the main text and tightly focused on what we did, that is the remote transport of high-dimensional OAM states without direct transmission of information carriers. Which have some similar features with the applications of dispersion-engineered sum-frequency generation in the field of time-frequency mode selection and measurement [Opt. Express 19, 13770 (2011), Phys. Rev. A 90, 030302 (2014), PRL 111, 153602 (2013)]. We have also cited these relevant references in our revision.

Furthermore, due to high-dimensional OAM states are encoded on the coherent beams, leading that therein multiple copies of information. Accordingly, comparing the transport fidelities with the classical bound, which is routinely used in the quantum teleportation, would also inappropriate. For this, we have also removed the comparison with classical bound.

Dear reviewer, thank you very much for these insightful and constructive comments. We have tried our best to consider your suggestions and rephrase the manuscript accordingly, which has clearly and exactly reflected our work. The main changes are as following.

In the section of Abstract.

We have rewritten to reflect what we truly did (See Lines 9-24 in Pages 1-2)

“The efficient transport and engineering photonic orbital angular momentum (OAM) lie at the heart of various related classical and quantum applications. Here, by leveraging the spatial-mode-engineered frequency conversion, we realize the remote transport of high-dimensional orbital angular momentum (OAM) states between two distant parties without direct transmission of information carriers. We exploit “perfect vortices” for preparing high-dimensional yet maximal OAM entanglement. Based on nonlinear sum-frequency generation working with a strong coherent wave packet and a single photon, we conduct the Bell-like state measurements for high-dimensional

“perfect vortices”. We experimentally achieve the average transport fidelity 0.879 ± 0.048 and 0.796 ± 0.066 for a complete set of 3-dimensional and 5-dimensional OAM mutually unbiased bases, respectively. Furthermore, by exploring the full transverse entanglement, we construct another strategy of quantum imaging with interaction-free light. It is expected that, with the future advances in nonlinear frequency conversion, our scheme will pave the way for realizing truly secure high-dimensional quantum teleportation in the upcoming quantum network.”

In the section of Introduction.

We have removed connections with teleportation and merely focused on the transport of OAM and nonlinear frequency conversion (See Lines 25-61 in Pages 3-4)

“Since the seminal discovery by Allen *et al.*¹, orbital angular momentum (OAM) of light has aroused numerous attentions as its high-dimensional nature provides the attractive potentials in some classical and quantum information protocols²⁻⁵, e.g., realizing large-alphabet optical communication^{6,7} and constructing the high-dimensional photonic quantum entanglement⁸, as well as provides the toolkit for the investigation of quantum fundamental questions⁹. However, these benefits are only conceivable when we are able to efficiently generate, manipulate, transport, and detect the OAM state. Hitherto, extensive investigations of OAM transport^{6,7,10-17} have been reported, such as OAM transport in free-space, specialized optical fiber, and even the distribution of OAM entanglement. Whereas, it is noted that the most of these previous schemes focused on the direct information transport only.

Recent years have also witnessed a growing interest in the nonlinear frequency conversion of the structured light^{5,18}. In spatial domain, it was demonstrated that using OAM light as fundamental frequency light source, one can realize the image processing and mode detection during the process of upconversion imaging^{19,20}, and by shaping the spatial shape of the pump source, one can enhance the performance of high-dimensional OAM frequency interface²¹⁻²⁴. In time-frequency domain, by using the dispersion-engineered sum-frequency generation, the quantum pulse gate for harnessing and selecting the time-frequency mode has been realized^{25,26}. Also, it has

provided a powerful tool for time-bin entanglement measurement²⁷. Particularly, we note that there have several attempts, by means of sum-frequency generation, to perform the coherent transfer of the polarization state between two distant parties without direct transmission of the information²⁸, and to realize the entanglement swapping of time-bin entanglement^{29,30}, however, the scenario of high-dimensional OAM state has not yet been fully explored. Some interesting questions arise naturally: whether the nonlinear frequency conversion can be used to transport the high-dimensional OAM state with an unprecedented way, and what new features this might lead to. These also form the major incentive of our present work.

Here, by leveraging the spatial-mode-engineered frequency conversion, we realize the remote transport of high-dimensional orbital angular momentum (OAM) states at a distance without direct transmission of information carriers. Specifically, by exploiting perfect vortices, we prepare high-dimensional yet maximally entangled orbital angular momentum (OAM) states as communication channel. And then, we employ sum-frequency generation (SFG) with a strong coherent state, which is used to bear the high-dimensional OAM state, to enhance the frequency conversion, and thus performing the perfect-vortex-based high-dimensional Bell-like state measurement reliably. Furthermore, we report the quantum imaging with truly interaction-free light.”

In the section of Discussion.

We discuss that our scheme could be extended to single photon scenario with the future advances in nonlinear frequency conversion (See Lines 283-302 in Pages 16-17)

“Here, we use a strong coherent light instead of single photons as the input state to enhance the SFG conversion (~ 4%) to perform the HDBLSM, and thus realizing the remote transport of high-dimensional OAM states from a coherent beam to a single photon. Therein the knowledge of the coherent source is not used in our scheme, which is similar with the one of key features of teleportation. However, in the original proposal of quantum teleportation, the input unknown state to be teleported should be encoded with single photon. If we adopt the single photon as input source, limited by the rather low frequency conversion efficiency, we need to wait a much longer time for

teleportation to occur. In this regard, it is expected that, with the future advances in structured light nonlinear frequency conversion, our scheme can work in the single photon scenario and will pave the way for realizing truly secure high-dimensional quantum teleportation in the upcoming quantum network.

In our scheme, Alice only needs to send the HDBLSM result via a classical link while the photon c received by Bob is always in the single-photon state. Then Bob is required to conduct a desired unitary operation on photon c , according to the results of HDBLSM (indicated by Eq. (2)), to recover the high-dimensional OAM states correctly. Thus, under ensuring the eavesdropper has no access to the coherent photons a , such a procedure can offer an additional security for the information transport and ghost imaging protocol. Whereas, in the direct transport of OAM states, the eavesdropper can acquire the information at any position along the communication channel.”

Comment 4. In addition, significant changes need to be made regarding the use of the technique for the transfer of optical images. Conceptually, introducing optical image transport as an implementation related to quantum imaging or teleportation-based is misleading. The use of these terms indicates the coherent transfer of information. Contrary to the proposal for quantum teleportation of a multimode field [Opt. Comm. 193 175 (2001)], the transferred information in the present experiment is not a spatially encoded quantum state, but the intensity information of an optical image encoded in a transverse spatial basis. The use of a non-standard measure of “image fidelity”, which relies only on intensity measurements of the prepared and measured image, neglects all phase information and cannot be compared to any quantum state fidelity or any non-classical bound. Furthermore, deductions of the capacity of a quantum channel (i.e., a channel that supports the coherent transport of quantum information) from these measurements make no sense because of the lack of evidence for coherence. The results only support the transfer of the amplitude of the encoded image, and thus, any claim of teleportation-based quantum transport is invalid and needs to be removed from the manuscript.

Response. Thank you very much for this insightful comment. It’s correct for the

reviewer to point out that “the use of a non-standard measure of “image fidelity”, which relies only on intensity measurements of the prepared and measured image, neglects all phase information and cannot be compared to any quantum state fidelity or any non-classical bound”. Actually, due to exactly acquiring the phase distribution of images is not easy in ghost imaging, and thus calculating the fidelities of ghost images remain challenging. For characterizing the performance of ghost imaging, the contrast-to-noise ratio (CNR) is a potential candidate [PRA 86, 063817 (2012); PRL 122, 123901 (2019)]. For clarifying this point, we have removed all claims of image fidelity and capacity of a quantum channel, and adopted the CNR to quantitatively characterizing the performance of ghost imaging of amplitude image, as following. (See Lines 265-281 in Pages 15-16)

“We present our experimental observations of the interaction-free-light-based ghost imaging of three-leaf and four-leaf Clover images in Fig. 5a and 5b, respectively. As shown by the dotted lines, we define the regions of interest (ROI) for both three-leaf and four-leaf Clover images, which occupy about 432 and 441 pixels, respectively. Here, for a quantitative analysis, we adopt the contrast-to-noise ratio⁴⁷, $CNR = (\langle G_{in} \rangle - \langle G_{out} \rangle) / \sqrt{\sigma_{in}^2 + \sigma_{out}^2}$ to characterize the image quality of recorded images, where $\langle G_{in} \rangle$ and $\langle G_{out} \rangle$ are the ensemble average of the photon numbers at any pixel inside and outside the ROI, respectively, while σ_{in}^2 and σ_{out}^2 are the corresponding variances. We have $CNR = 1.37$ and 1.58 for three-leaf and four-leaf Clover images, respectively, which are at the same level with the traditional ghost imaging^{46,48}, therefore confirming the good performance of our system. Noted that, the edges of recorded images in Fig. 5a and 5b are not sharp compared with the original ones (indicated by insets), which mainly caused by both the non-maximal spatial entanglement for photon pairs generated via SPDC and the nonuniform SFG efficiency for each pixel in the HDBLSM stage. Also, a better time and spatial overlap of all the image planes of the ICCD camera, BBO crystal, and the SLM1 can further improve the CNR.”

Fig. 1. (a1-a5) phase structures formed by interfering between plane wave and perfect vortices with $\ell = -2$ to $+2$, respectively. (b1-b5) Experimental observation of doughnut-like intensity patterns for these perfect vortices.

As for the coherent transfer of image, actually, the OAM state can be seen as a special amplitude-phase image in 2D space. Which is because that OAM mode is a typical structured mode in spatial domain. As shown in Fig. 1, the spatial-varied phase and amplitude (or intensity) can be seen clearly. In this regard, the good performance of the transport of high-dimensional OAM states can also reflect the capacity of image coherent transfer. Whereas, due to the fact that there doesn't exist a universal solution for calculating the fidelity of ghost imaging, thus here, we merely state that ghost imaging of amplitude image. Besides, as inspired by both your concerns and also the comments by Reviewer #1, we clearly reveal the connection between our scheme and a new variant of ghost imaging protocol, that is ghost imaging with interaction-free light. We have clarified this point in our revision, as following. (See Lines 226-236 in Pages 13-14)

“...Actually, the aforementioned three- and five-dimensional OAM states can also be seen as the 2D complex images in spatial domain. In this regard, our scheme can also represent a new variant of ghost imaging. In the conventional ghost imaging, it necessitates two correlated beams; one beam illuminates an object while the image is recovered from the other beam that has never interacted with the object⁴⁴. However, these two beams actually share a common past, i.e., created from the same pump via SPDC. In contrast, our scheme uses the coherent state a to bear the image information, and then transport nonlocally to photon c after HDBLSM is done. Obviously, the

photons used for object illumination and for image recovery do not share any common past, i.e., truly interaction-free. It's noted that there have also several attempts, by using entanglement-swapped photons, to achieve the ghost imaging⁴⁵....”

“...Further, by exploring the full transverse spatial mode entanglement, we have succeeded in realizing the ghost imaging of amplitude objects with interaction-free light...” (See Lines 309-311 in Page 17)

Comment 5. This manuscript can be of significance to the field because it demonstrates a technique for arbitrary projections of OAM states in up to five dimensions, allowing for the coherent transfer of spatial information. Nevertheless, I can only recommend the publication if the previous concerns are addressed, and the relevant changes are implemented.

Response. Thank you very much again for positive recommendations and insightful suggestions. We have followed all of your suggestions to revise our manuscript. Therein tightly focusing on what we have really done, i.e., remote transport of high-dimensional OAM states from a coherent light beam to a single photon, and without referring to the traditional quantum teleportation. By addressing your concerns, we indeed feel that our manuscript has been much improved now, which we sincerely wish you could consider in Nature Communications.

REVIEWERS' COMMENTS

Reviewer #2 (Remarks to the Author):

I've now gone through the response and new version of the manuscript, where the authors have implemented important changes to exactly represent what they have done. By changing the title and modifying the general narrative, they have addressed my main concerns. Furthermore, they have removed and/or corrected the technical problems I pointed out on their results.

With this, I confirm that the manuscript has significantly improved and fulfils the requirements for publication in Nature Communications.